# Self-sustained biphasic catalytic particle turbulence

Ziqi Wang [1,3], Varghese Mathai [2,3] & Chao Sun [1]

Turbulence is known for its ability to vigorously mix fluid and transport heat. Despite over a century of research for enhancing heat transport, few have exceeded the inherent limits posed by turbulent-mixing. Here we have conceptualized a kind of "active particle" turbulence, which far exceeds the limits of classical thermal turbulence. By adding a minute concentration ($\phi_v \sim 1\%$) of a heavy liquid (hydrofluoroether) to a water-based turbulent convection system, a remarkably efficient biphasic dynamics is born, which supersedes turbulent heat transport by up to 500%. The system operates on a self-sustained dynamically equilibrated cycle of a "catalyst-like" species, and exploits several heat-carrier agents including pseudo-turbulence, latent heat and bidirectional wake capture. We find that the heat transfer enhancement is dominated by the kinematics of the active elements and their induced-agitation. The present finding opens the door towards the establishment of tunable, ultra-high efficiency heat transfer/mixing systems.

---

[1] Center for Combustion Energy, Key Laboratory for Thermal Science and Power Engineering of Ministry of Education, Department of Energy and Power Engineering, Tsinghua University, 100084 Beijing, China. [2] School of Engineering, Brown University, Providence, RI 02912, USA. [3] These authors contributed equally: Ziqi Wang, Varghese Mathai. Correspondence and requests for materials should be addressed to C.S. (email: chaosun@tsinghua.edu.cn)

Turbulence is ubiquitous in nature and industry[1–4]. In the global context of energy exchange, thermal turbulence holds a central place as a major driver in many natural and industrial processes[5–8]. The primary heat transporters of thermal turbulence are the "plumes" (a form of coherent turbulent structure)[6], and they form an essential component in ocean currents[9,10], atmospheric and mantle convection[11], volcanic eruptions[12], biochemical and combustion reactions[13], as well as in the sustenance of thermonuclear phenomena in the sun, the stars, and other galactic powerhouses[14,15]. A remarkable property of these coherent plume structures is their ability to mix initially distinct constituents across multi-decadal length and time scales, yielding extremely efficient (super-diffusive) transport of scalars[16], which are typically orders of magnitude faster than those achievable by molecular diffusion alone. The same characteristics of the plumes have become to be exploited in design of efficient heat exchangers and mixing networks, and, today, accounts for roughly 80% of the world's heating, ventilation, and air-conditioning (HVAC) load[17,18].

From a fundamental perspective, a central question for any heating or cooling device is to establish a robust relationship between an applied temperature difference and the corresponding heat flux[6]. For thermal turbulence, this can be expressed in dimensionless form as a relation between Nusselt number Nu (or the dimensionless heat flux) and Rayleigh number Ra (or the dimensionless temperature difference), with an effective scaling law[5]: $\mathrm{Nu} \propto \mathrm{Ra}^{\beta}$. Here, $\mathrm{Nu} = Q/\left(\lambda \frac{\Delta T}{H}\right)$, $\mathrm{Ra} = g\gamma\Delta T H^3/\nu\kappa$, where $Q$ is the measured heat input through the bottom plate into the system per unit time, $\lambda$ the thermal conductivity of the working fluid, $\Delta T$ the temperature difference, $H$ the thickness of the working fluid layer, $g$ the gravitational acceleration, $\gamma$ the isobaric thermal expansion coefficient, $\nu$ the kinematic viscosity, $\kappa$ the thermal diffusivity, and $\beta$ the effective scaling exponent. This power law dependence can be attributed to the interaction between the turbulent bulk flow and boundary layer[19]. However, in classical thermal turbulence[20], the Nusselt number scales with Ra by an effective scaling exponent $\beta \le 1/3$. This weak scaling underlies a vast number of natural and engineering processes around us, and therefore presents a significant limitation to heating/cooling devices spanning a broad spectrum of applications from chemical and bio-reactor technologies to electronics and power engineering.

In recent years, there has been a rapid surge in the number of studies aimed at furthering the heat exchange capabilities using convective turbulence. In turbulent convection, this has been achieved by controlling the coherence of plumes using either surface treatment (wall roughness: symmetric or asymmetric or fractal[21–23]), or through confinement[24,25]. Others have tapped in on modifying the working liquid's properties by introducing additives, including polymers, (plasmonic) nanoparticles and nano-emulsions[26,27], and more recently through thermally responsive particles[28]. Some of these have been successful in attaining heat flux enhancements essentially through an increase in the prefactor of the Nu–Ra scaling within the classical regime of thermal turbulence. The above modifications have at best resulted in around 50% enhancement, yet with a profound impact in many heating and cooling applications. Injecting gas bubbles[29], or boiling the working liquid to the point that vapor bubbles[30] form can also yield heat flux enhancement[31–33]. These approaches are feasible for open systems (for gas escape), but require additional energy input (for gas injection) and/or extremes in operating temperatures (to boil). Hence, these are of limited applicability for the widely prevalent closed-system heat exchangers[34].

Strikingly, here we show that introducing a minute volume fraction of a hydrofluoroether liquid (HFE-7000) to a classical turbulent convection system based on water, the heat transport can enhance up to 500%. We explain the underlying mechanism of heat transfer enhancement and find that it is dominated by the kinematics of the active elements and their induced-agitation.

## Results

**Heat transfer enhancement**. The recent surge in explorations aimed at marginal heat flux enhancements in thermal convection underlines the need for fundamentally new concepts of heat transport in closed-system heat exchangers. One may ask the question: is there an elegant alternative to achieve significant heat transfer enhancements with minimal modifications to a closed system? We begin with a classical (water-based) thermal convection system (aspect ratio 0.5; see also Suppl. Mat.), with a heated bottom plate ($T_b \approx 35\,°C$) and a cooled top plate ($T_t \approx 5\,°C$) at atmospheric pressure $p_0$. To this, we introduce a minute fraction ($\phi_v \sim 1\%$) of a low conductivity liquid (hydrofluoroether (HFE-7000)), through which a highly efficient active particle turbulence is born (Fig. 1a and Suppl. Mat.). The active species enters a dynamic boiling condensation cycle, thereby creating biphasic heat-carrier elements (see Fig. 1b–d) and supplanting the plumes (classical coherent structures) of thermal turbulence (Fig. 1e, f). The collective dynamics of the vapor-liquid elements gives rise to even stronger coherent structures than in thermal turbulence. We report heat flux enhancements of up to 500% (green-shaded half of Fig. 1g).

When the bottom plate temperature $T_b$ is raised above a critical value $T_{cr}$, the heat transfer ($\mathrm{Nu}/\mathrm{Nu}_0$) increases almost linearly (Fig. 1g), where $\mathrm{Nu}_0$ is the heat exchange possible due to convective turbulence alone. This occurs as a consequence of increased activity of the biphasic particles (see insets Fig. 1b–d). Remarkably, the fivefold enhancement in heat flux is attained with only a 1.3% (by volume) activity of the biphasic species (inset to Fig. 1g). The HFE-7000 liquid boils to form vapor bubbles, and the vapor bubbles detach from the bottom plate and rise due to their buoyancy. During their rise through the bulk of the cell the vapor partially condenses into liquid; so the bubbles become biphasic with increasing liquid fraction, and consequently decreasing in buoyancy. These biphasic particles, upon reaching the top plate fully condense into liquid droplets, and then fall back onto the bottom plate. This completes a so-called dynamically equilibrated cycle of the biphasic species. Next, we highlight three identifiable levels in this heat enhancement process (Fig. 2a–c). At $T_b < T_{cr}$ the system is in a state of purely plume-driven thermal turbulence ($\mathrm{Ra} \approx 4.5 \times 10^{10}$) with also a large-scale circulation, the HFE-7000 liquid layer spreads on the bottom plate (see Fig. 2a). Upon raising the bottom plate temperature to $T_b - T_{cr} > 0\,K$ the additive species (HFE-7000) gains in activity (Fig. 2b, c). In the partially active regime (Fig. 2b), biphasic plumes pinch off from the bottom plate (see also Supplementary Movie 6) with only a fraction of the HFE-7000 species taking part in the activity. In the fully active regime (Fig. 2c), the bottom plate is cleansed of the HFE-7000 liquid layer, and the flow develops a coherent azimuthal rotation along the vertical direction, hereafter referred to as the Sweeping mode (see Supplementary Movie 7). Until at $T_b - T_{cr} \approx 10\,K$ we observe strong active particle transport with $\mathrm{Nu}/\mathrm{Nu}_0$ approaching a value of 6. An obvious consequence of this biphasic activity can be seen in the temperature time-series (insets (data plots): Fig. 2a–c) measured in the bulk of the set-up (see also Suppl. Mat.). The low frequency, large amplitude temperature fluctuations (inset (data plot) to Fig. 2a), signatory of the thermal plumes, are replaced by a higher frequency but lower amplitude temperature fluctuations at the intermediate level of biphasic activity (inset (data plot) to Fig. 2b). However, upon increasing $T_b - T_{cr}$ further, we attain a

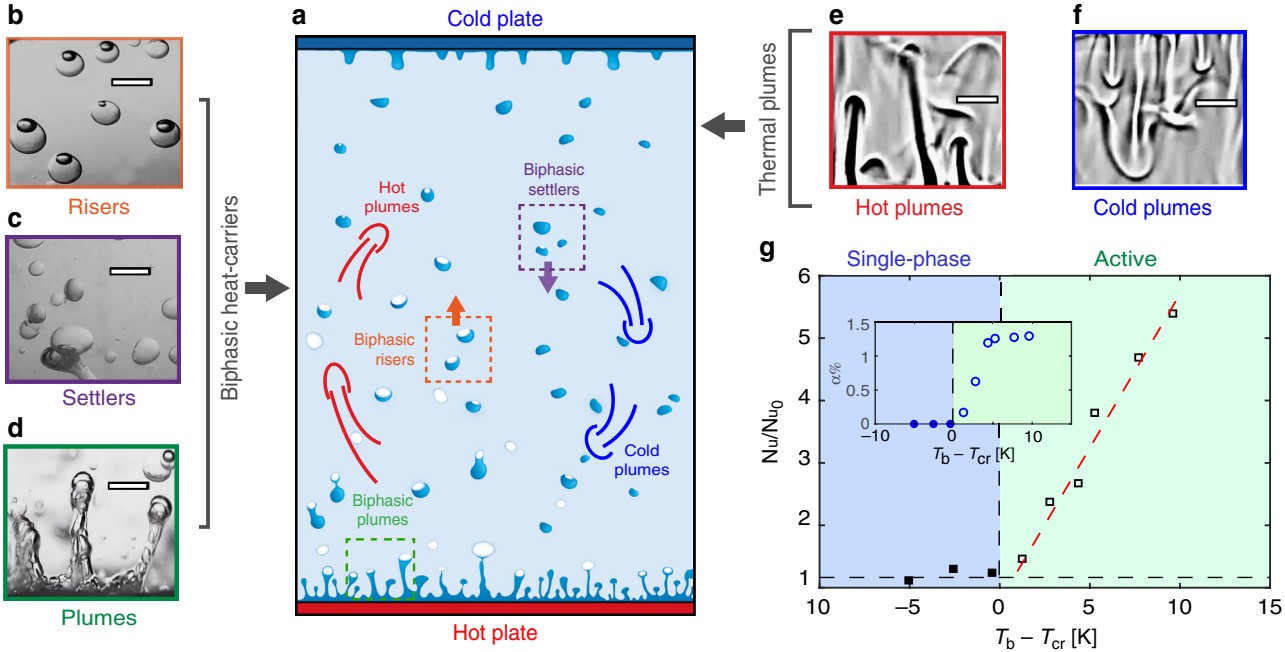

**Fig. 1** Active biphasic particle turbulence. **a** Schematic of a classical thermal turbulence set-up with water as working liquid, and augmented by ~1% volume of active species: HFE-7000. Insets **b**–**d** show novel heat carriers of the system: **b** biphasic risers, **c** settlers, and **d** plumes (see Supplementary Movie 1 and 2). Insets **e**, **f** show shadowgraphy images of the classical heat transporters of thermal turbulence, namely **e** hot and **f** cold thermal plumes (see Supplementary Movie 3). Inset scale bars: 6 mm. **g** Heat flux enhancement expressed as the ratio of Nusselt numbers $Nu/Nu_0$ vs. temperature difference $T_b - T_{cr}$. Here, $Nu$ is the Nusselt number of the current system, $Nu_0$ is the Nusselt number of classical thermal turbulence, $T_b$ is the bottom plate temperature, and $T_{cr}$ is the boiling point of HFE-7000 liquid above which the dynamic evaporation-condensation cycles begin. The blue shaded area corresponds to the single phase regime, almost identical to classical thermal turbulence, and the green-shaded area the active regime. Inset shows the vapor phase of the biphasic species volume fraction $\alpha$ (see Suppl. Mat.) as function of $T_b - T_{cr}$. The experiments were conducted at fixed bottom and top plate temperature difference $\Delta T \equiv T_b - T_t$ in order to maintain Rayleigh number almost unchanged

state of intermediate frequency and amplitude temperature fluctuations (inset (data plot) to Fig. 2c), which optimally enhances the heat transport (see Suppl. Mat. for details). This non-monotonic trend of the temperature signal with increasing biphasic activity (Fig. 2a–c) suggests intricate underlying transport mechanisms. In order to gauge the particle dynamics, we superimpose Lagrangian trajectories of the active particles (Fig. 2d, f). At the intermediate activity levels, the biphasic species trajectories (Fig. 2d) are restricted to the lower half of the convection system; the inset reveals the detailed kinematics of such a biphasic particle. The vapor-phase volume is consumed during the particle's ascent, and it thereby sustains a velocity cycle for the particles (see Fig. 2e). The flow is highly dynamic, but the collective motions of the flow and the global responses of the system are stable over time. In other words the ratio of liquid and vapor phases change locally and dynamically, but substances move between the states at an equal rate, meaning there is no net change in time. In contrast, at the optimal level of activity, we observe complete up-down travel of the vapor-liquid elements (Fig. 2f). As will be shown later, the large-scale circulation (LSC) set-up by the collective motion of the particles far-exceeds the LSC velocities $v_f \approx 0.2(\nu/H)\sqrt{Ra/Pr}$ of classical thermal turbulence. What is truly remarkable is that these processes are self-sustained. The biphasic particles appear from the bottom plate with a lower density (positive buoyancy) as compared to water, and after releasing heat in the bulk and near the top plate, they increase their overall density (gain negative buoyancy) and move downward to recover their initial state. Thus, the role of the biphasic species to the thermal convection system holds a close analogy to that of a "catalyst".

**Mechanism of heat transfer enhancement**. Next, we assess the detailed activities of mixing and heat transport by the active biphasic particles. To visualize this, we release a passive scalar (Rhodamine B dye, see Suppl. Mat. for details) into the system. The dye is illuminated using a pulse laser with pulse duration of 7 ns (Vlite-200 532 nm Solid State Laser System). A camera that is used for capturing the mixing process is attached with a high-pass (green) filter to mask stray reflections from the particles (see Suppl. Mat.). Figure 3a, b show image sequences where the background turbulence advects and mixes the fluorescent dye in classical thermal convection and biphasic particle-laden turbulence, respectively. In the classical system, the dye mixes along the path of the LSC (see Fig. 3a and Supplementary Movie 10). In contrast, the biphasic system displays fast and chaotic mixing (Fig. 3b) over a wide range of length scales (see also Supplementary Movie 11 and Suppl. Mat. for details). We measured the collective LSC velocity $V_c$ (Fig. 3c) in these systems. In a state of classical thermal turbulence, $V_c \approx 2$ cm s$^{-1}$, i.e., comparable to the free fall velocity scale with corrections ($v_f = 0.2\nu/H\sqrt{RaPr} \approx 3.6$ cm s$^{-1}$) for buoyancy driven convection. However, when the bottom plate temperature is raised 10 K above $T_{cr}$, we observe a dramatic 12.5-fold increase in $V_c$ (from 2 cm s$^{-1}$ to 25 cm s$^{-1}$) in the active particle system. Again, the beauty of the process is that these active particles are collectively self-sustained. As compared to the self-sustained three-way coupled interactions between the transport modes associated with fluid flow, particles, and radiation[35], here we use the phase transition cycle and its coupled effects with biphasic particles motion and carrier flow.

The effective heat transport due to this collective motion of biphasic particles can be written as $Q_l = \frac{\pi d^2 \rho_v}{4} \mathcal{L} V_c \alpha$, where $\rho_v$ is

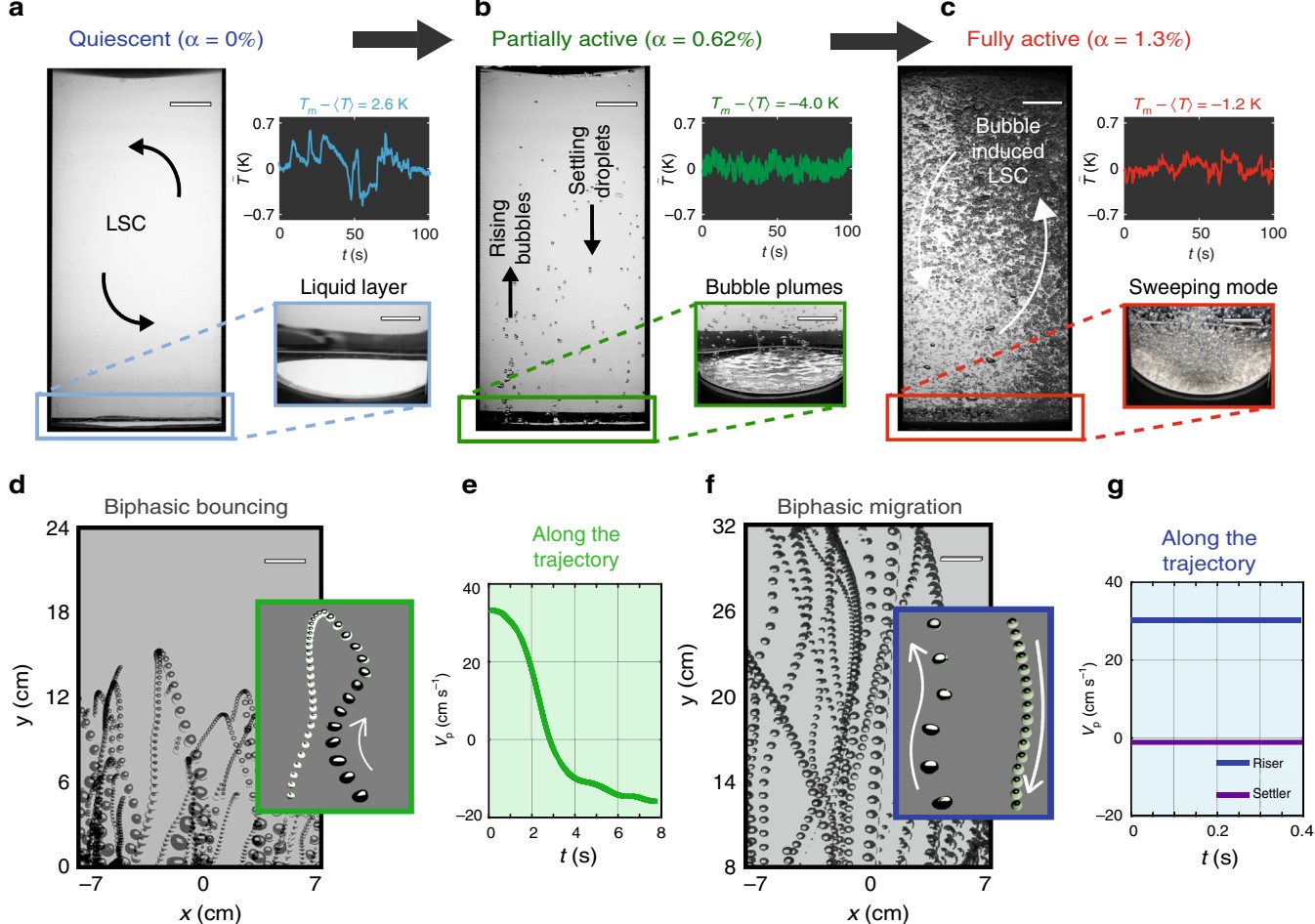

**Fig. 2** Representative stages of heat exchange process. Three identifiable stages revealing increasing levels of biphasic particle activity (see Supplementary Movies 4 and 5), namely **a** single-phase regime ($T_b - T_{cr} < 0$ K), **b** partially active regime (0 K $< T_b - T_{cr} < 5$ K), and **c** fully active regime ($T_b - T_{cr} > 5$ K). Insets (snapshots) of **a–c** show the zoom-in views of the bottom plate (see Supplementary Movies 6 and 7). Insets (data plots) of Fig. 2**a–c** show temperature time-series measured in the bulk of the set-up. $T_m = (T_b + T_t)/2$ is the system mean temperature, $\bar{T}$ is the time-averaged temperature. **d** Biphasic bouncing regime observed under partially active condition (see Supplementary Movie 8). Inset (snapshot) reveals the detailed kinematics of a representative biphasic particle, with both particle size and velocity changing during its rise and fall. The particle velocity $V_p$ evolving with time is shown in **e**. **f** Biphasic migration regime, observed at a higher activity level. Here, the particles migrate from bottom plate to top plate and vice versa (see Supplementary Movie 9). Inset shows typical riser and settler trajectories. **g** Evolution of $V_p$ for riser (blue) and settler (purple) particles in time. The velocity shown in **e**, **g** is the vertical velocity, with positive denoting upward travel of the biphasic elements, and negative denoting downward motion. Scale bars: 50 mm in **a–c**; 25 mm in **d** and **f**

the density of vapor phase of HFE-7000, $\mathcal{L}$ the latent heat, $V_c$ the collective velocity, and $\alpha$ the effective volume fraction of the biphasic species. While this accounts for a part of the observed heat flux enhancement, i.e., $\mathrm{Nu}_l = Q_l / (\lambda \frac{\Delta T}{H})$ where $\mathrm{Nu}_l$ is the contribution of biphasic kinematics mechanism to the non-dimensional heat flux enhancement, its difference from the total heat flux enhancement is still finite. The local mixing observed in the dye-visualization experiments (Fig. 3a, b) suggests that the biphasic particles agitate the surrounding liquid, much as in the same way as rising air bubbles[36] induce liquid velocity fluctuations. To model this we estimate the typical buoyancy-based velocity scale $V_g \sim \sqrt{g d_{bi}(1 - \Gamma_{bi})}$, where $d_{bi}$ and $\Gamma_{bi}$ are the effective diameter and the effective density ratio[37], respectively, of the biphasic particles. The ratio of buoyancy to viscous forces $V_g d_{bi}/\nu \in [990, 1400] \gg 1$, and therefore, the wakes of the rising/settling biphasic particles are turbulent[38]. The passive scalar transport can be modeled as an effective diffusive process[39] and thus the Nusselt number can be expressed as $\kappa_e/\kappa$ with the effective diffusivity $\kappa_e \propto u'\Lambda$, where $\Lambda$ is the integral length scale

of the liquid velocity fluctuations[40] (see also Suppl. Mat.). Then the induced agitation[41] by the biphasic particles can be modeled as $u' \approx V_c\sqrt{\alpha}$. Plotting the differential heat flux enhancement $\delta\mathrm{Nu} - \mathrm{Nu}_l$ against $V_c\sqrt{\alpha}$, we again obtain a linear growth (Fig. 3d). Here, $\delta\mathrm{Nu} = \mathrm{Nu} - \mathrm{Nu}_0$ and $\mathrm{Nu}_l = Q_l / (\lambda \frac{\Delta T}{H})$. This provides further evidence that the induced agitation (in water) by the biphasic particles accounts for the remainder of the heat exchange, which is the major contributor. Thus, the effective heat transfer gain (up to 500%) of biphasic turbulence is believed to result from the combined contributions of (a) the kinematics of the active particles, and (b) their induced liquid agitation (Fig. 3e). Finally, we compare the active particle driven heat transport to the extrapolated Nu based on Nu vs Ra scaling of classical thermal turbulence[5] (see inset to Fig. 3e). To achieve a comparable heat flux enhancement through thermal plumes would demand a 65-fold increase in the effective Ra, i.e., equivalent to an effective $\Delta T$ increase from 30 to 1950 K, which is unrealistic for most industrial applications. In comparison, our biphasic particle system attains this state through minimal operational

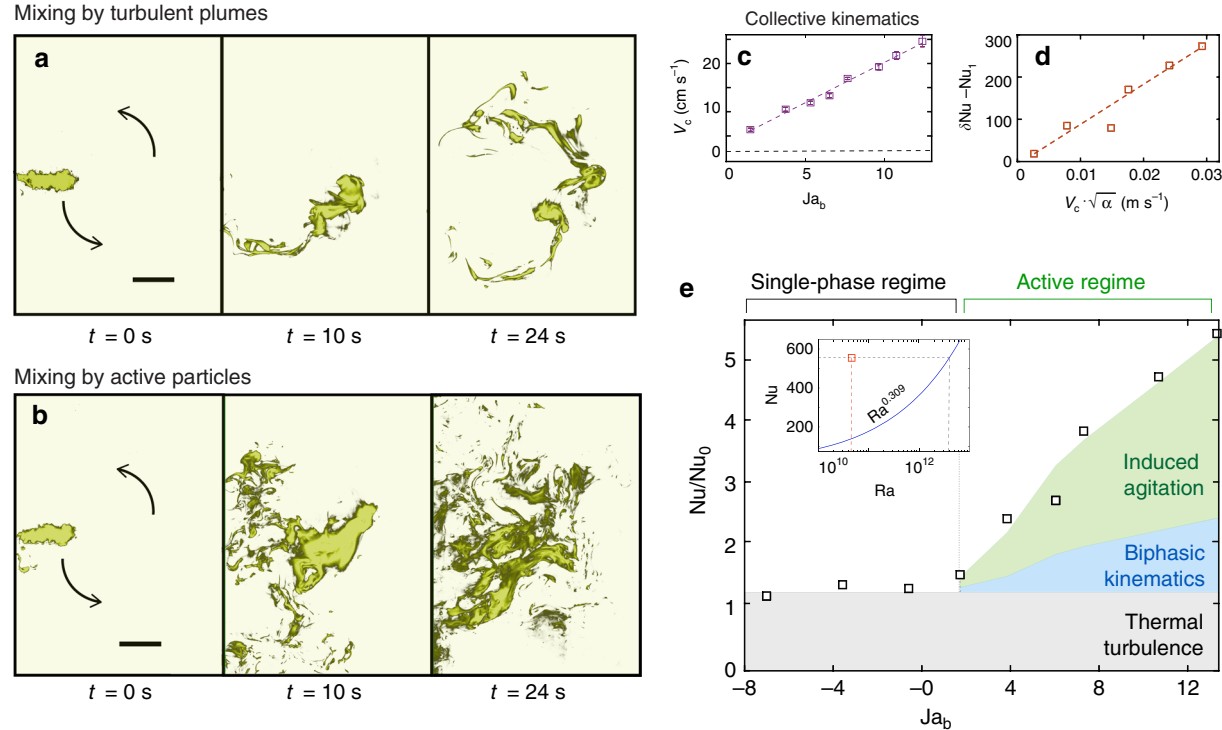

**Fig. 3** Modeling of mixing and heat transport mechanisms. Snapshots of mixing of a patch of fluorescent dye (passive scalar) released in **a** classical thermal turbulence system and **b** active biphasic turbulence, at time instants (left to right) 0, 10, and 24 s after injection. In classical thermal turbulence **a**, the spreading occurs primarily along path of the large-scale circulation (LSC), while in the active biphasic case **b**, the mixing is fast and occurs across a wide range of scales (see Supplementary Movie 10 and 11 for the comparison). False coloring was adopted based on the intensity of the dye. **c** Collective LSC velocity $V_c$ in the active regime vs. Jakob number $Ja_b$ of the bottom plate. The LSC velocity for classical thermal turbulence (black dashed line) is also shown for comparison. **d** Differential heat flux $\delta Nu - Nu_l$ vs. theoretical estimate of liquid agitation[39] $u' = V_c\sqrt{\alpha}$. Here, $\delta Nu$ is the total heat transfer enhancement, $Nu_l$ is heat transfer enhancement contributed by biphasic kinematics, and $u'$ is the theoretically estimated liquid velocity fluctuation (r.m.s) based on the measured $V_c$ and volume fraction $\alpha$. **e** Nusselt number for increasing Jakob number $Ja_b$ (see also Suppl. Mat.). The shaded areas show different contributions to the total heat transfer enhancement, namely thermal turbulence (gray), biphasic particle kinematics (blue) and induced liquid agitation (green). The inset compares active particle induced heat transport with the Nu vs. Ra scaling of classical thermal turbulence. Scale bar: 22 mm

modifications to the existing thermal turbulence system, which may revolutionize future industrial designs.

## Discussion

In summary, we have created an efficient class of active biphasic thermal turbulence, which provides sustained heat transport enhancements (by up to 500%) that well-surpasses the limits achievable by classical thermal turbulence. The biphasic "activity" is born through addition of a minute quantity (~1%) of a low conductivity, engineered liquid (HFE-7000) to a thermal convection set-up. This active species is able to extract energy from very small temperature gradients through their evaporation-condensation cycles. This, when coupled with their kinematics introduces several efficient heat-carrier mechanisms within a single system, leading to a fivefold enhancement in the heat transport. Remarkably, the system shows regimes of linear scalings of heat exchange with temperature change (for independent changes in bottom and top plate temperatures), and for aspect ratio changes (see Suppl. Mat. and Supplementary Movie 12), suggesting robust predictability and heat transfer control in both heating and cooling applications. When compared to other active matter systems (Janus particles[42], Kinesin rods[43], plasmonic nanoparticles[44], etc), the biphasic transport here is efficient, self-sustained and dynamically equilibrated, and requires minimal operational modifications to existing closed-system heat exchangers[34]. Furthermore, the biphasic species (HFE-7000) is non-corrosive, non-flammable, and non-ozone depleting (ODP

0), and hence holds promise as a safe and highly effective "catalyst-like" additive in contemporary clinical, biochemical, and nuclear engineering settings. Finally, the fast chaotic mixing by the active particles (Fig. 3b) extends the paradigm of applications towards vigorous (low-shear) mixing devices that are devoid of moving elements, desirable in biologically active environments. Finally, we note that the current work has only uncovered a subset of the rich possibilities of active biphasic turbulence in terms of the parameter space. In a future investigation, we plan to map out the system behavior under varied settings of ambient pressure, volume fraction and Rayleigh numbers, as well as within flow networks.

## Methods

**Experimental set-up and experiments**. The experiments were performed in the classical (water-based) thermal convection system to which we introduce a minute concentration of HFE-7000. The working fluid is confined between a copper top plate, which is cooled by a water-circulating bath (PolyScience PP15R-40), and a copper bottom plate, which is heated by the Kapton film heater. During the experiment, the working fluid inside the experimental system will induce thermal expansion and biphasic activity, and therefore a small open reservoir is connected to the cell, which plays a role as expansion vessel to compensate the volume change. Prior to the experiments, water and HFE-7000 were degassed so that all dissolved air can be removed (see Suppl. Mat. for details). We did overall heat flux measurements and dye mixing visualization experiments. When conducting heat flux measurements, the top plate is maintained at constant temperature and the bottom plate is heated through a constant heat flux. We use a PID (Proportional-Integral-Derivative) controller to regulate the temperature of different parts of the set-up. All the experiments were performed under atmospheric pressure so that the critical temperature $T_{cr}$, at which biphasic activity begins, remains unchanged. To

visualize the mixing induced by the biphasic elements, we release a dye (Rhodamine B) into the system. The Rhodamine dye is available in powder form, with a small particle size and a high Schmidt number[45], so that it can be treated as a passive scalar in the flow. A pulse laser (Vlite-200 532 nm Solid State Laser System) to illuminate the dye. A camera (HiSense Zyla camera from DANTEC DYNAMICS), attached with a high-pass (green) filter to mask stray reflections from the particles, is used to record the images (see Suppl. Mat. for details).

**Working fluid**. The working fluid is deionized ultrapure water. We introduce a minute concentration (~1%) of low-conductivity heavy liquid called HFE-7000 liquid (manufactured by 3M$^{TM}$). HFE-7000 (1-methoxyheptafluoropropane) here acts as an "additive catalyst", and is almost immiscible (60 ppmw) in water (see Suppl. Mat. for details). It currently has a wide variety of uses in pharmaceutical, chemical, electronics applications and so on.

**Calculation of biphasic volume fraction**. The method we use to get the volume fraction $\alpha$ is as follows. We measure the water surface height $h$ inside the expansion vessel and according to $h$ we can calculate $\alpha$ based on the conservation of mass and conservation of volume. The expansion vessel is with scales from which we can read the water surface height $h$. The scale on the expansion vessel has the minimum scale value 1 mm, which corresponds to 0.007% of $\alpha$. As the bottom plate temperature increases, there are two regimes, namely single phase and active particle convection regime. In the single phase regime, the working liquid goes through purely isobaric thermal expansion, and in the active particle convection regime the bottom plate temperature is above $T_{cr}$, so the biphasic activity begins, which induces volume change as well. No matter how much the total volume changes, the total mass remains constant. For each case, after the system reaches the finally statistically stable state, we then conduct about 4-h heat flux measurements. During the heat flux measurements, we monitor $h$ from the expansion vessel scales every other 30 min and we in total obtain eight readings of $h$. On top of this, the minimum and maximum variation in $\alpha$ during the four hours measurement was 1.3% variance of $\alpha$. The value $h$ we use to calculate $\alpha$ is the mean value of the eight readings. (see Suppl. Mat. for details).

**Image processing**. Images of biphasic bouncing and biphasic migration are generated by superimposing several particle trajectories through MATLAB processing. The collective velocity is analyzed using PIV (Particle Image Velocimetry). More processing details can be found in Suppl. Mat.

## Data availability
The data that support the findings of this study are available from the corresponding author upon reasonable request.

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

## Acknowledgements
We thank D. Lohse, A. Prosperetti, S. Maheshwari, and X.-F. Xu for useful discussions. This work was supported by Natural Science Foundation of China under grant no. 91852202, 11861131005, 11672156.

## Author contributions

C.S. conceived the project in 2015 during the CISM course in Udine, Italy. C.S., Z.W. and V.M. designed the research. Z.W. and V.M. performed the experiments and analyses. V.M., C.S. and Z.W. wrote the manuscript. Z.W. and V.M. contributed equally to this work.

## Additional information

**Competing interests:** The authors declare no competing interests.

