## [peer review file · Nature Communications]

Reviewer #1 (Remarks to the Author):

Please see attached review comments.

Reviewer #2 (Remarks to the Author):

The manuscript presents an experimental study of a Rayleigh-Bernard convection with water and a small addition of HFE 7000 liquid. Under different temperature of the bottom plate, the HFE 7000 vaporises differently and generates bubbles, which the authors call active particles. They show a 5-fold increase in heat transfer than conventional convective heat transfer. It is concluded that the increased heat transfer is due to both the kinematics of the bubble and the bubble induced turbulent flow. The experimental work is nicely done, with the flow visualization, bubble velocity calculation, dye mixing measurements, etc. The work should be reproducible giving the level of details provided.

The idea of inducing turbulence/bubbles to increase the heat transfer rate has been spelled previously. Surprisingly, there are very few experiments with multiphase (two-fluid mixture) Rayleigh Bernard convection. Though the HFE7000 is a known thermal fluid, its application to the RBC system is new. The present work is novel and should be of interest to a wide range of audiences interested in heat exchange.

Overall, I think the paper is interesting and is suitable for publications in Nature Communications. Below are several questions to improve the presentation of the results.

The authors emphasis in several places the self-sustainability of the particle (bubble) generated turbulence in the text. However, there are no experimental data showing, for example, long time behaviour of the system. Can the authors use some quantitative measure to show the sustainability?

In Fig.3e, is the boundary of the shaded region (biphase kinematics) calculated using eq.15-17 from the SI? Since this is one of the main conclusions of the paper, it would be nice to have eq.15 in the main text.

Reviewer #3 (Remarks to the Author):

This article reports on the experimental demonstration of a novel (or anyway not previously identified) class of multiphase turbulent transfer. This is the result of the addition of a small amount of heavy fluid to a classic convective cell containing water. The heavy fluid boils at relatively low temperature, producing buoyant "particles" that rise and settle following a dynamic condensation/evaporation equilibrium. The mixing induced by these particles produces a stunning five-fold increase of Nusselt number, compared to a classic turbulent thermal convection. The evidence provided by the authors is compelling and strongly supports their claims.

The result is truly remarkable, with potential impact in a vast range of applications from nuclear to biomedical engineering, as highlighted by the authors. Therefore this work is expected to be of interest to a broad scientific audience. Moreover, it surfaces a rich and relatively unexplored dynamics of self-sustained particle-driven turbulent convection. Thus the work will also trigger strong interest from the specialized community in the area of turbulent multiphase flows.

In the interest of clarity and completeness, I suggest the following points are addressed before publication:

- The authors refer to a "critical temperature" T_{cr} , beyond which the heavy fluid "activates". This is actually the boiling temperature of this fluid, but this is only clear in the supplementary material.

It should be clarified in the main article.

- Moreover, the “particles” of the heavy fluid are shown as “rising bubbles” and “falling droplets” in Figure 2, but one has to look carefully at such figure to realize that there is a complete phase transition involved. The authors need to be clearer in describing the steps of the process, illustrating how the fluid evaporates and condense, and clearly stating in which part of the dynamic process the fluid is gaseous/liquid.

- The five-fold increase of Nusselt number is impressive, and so is the twelve-fold increase in convection velocity scale. However, it is not obvious how those quantities might scale. The authors do provide interpretation/explanation on the processes leading to the enhancement of turbulent fluxes, but it is not evident how this can extrapolate to, for example, higher temperature differences. Is the linear behavior in Figure 1(b) expected to continue?

- Figure 2(c) is supposed to show “a state of high frequency and high amplitude temperature fluctuations”, compared to the other panels in the same figure. But from the figure I’d rather say that the fluctuations have intermediate frequency and amplitude, between those of panels (a) and (b). Maybe things would be clearer if the Supplementary Material included a Fourier analysis of those temperature fluctuations.

- I couldn’t find the definition of the variable T_m in Figure 2.

- The authors rightfully stress the novelty of their findings. One of the few examples of regimes in which particles are driving turbulent thermal convection in a self-sustaining fashion is the work of Zamansky et al., Phys Fluids 26, 071701, 2014. Similarity and differences with respect to the present case should be mentioned.

- The phrase “quiescent regime”, used by the authors to refer to the regime before the activation of the heavy fluid, is contradictory with the fact that this really is a turbulent convection. A different wording is in order.

May 23, 2019

1 Response to Reviewer 1

Synopsis: *The paper presents experimental results for self-sustained catalytic particles, which are claimed to enhance fluid mixing and convective heat transfer in an enclosed convection cell. The paper presents interesting results and arguments based on experimental analysis to suggest that the catalytic particles are capable of enhancing the overall heat transfer process through the kinematics of active particles and their induced liquid agitation. The paper is well written and presents convincing arguments for most observations; however, the paper can still be improved by addressing the concerns and comments shown below. In summary, the paper needs to be revised before it is accepted for publication. More detailed suggestions and review comments are as follows:*

Response: We thank the reviewer for the detailed reading of our manuscript and for his/her positive comments. Below we provide detailed answers to the questions raised by the reviewer. We also implemented the reviewer's suggestions into the paper.

Comment 1: *Introduction: The equation for Nusselt number shown in the paper is based on heat rate instead of heat flux, as it is routinely done in heat transfer publications. Why? Is the H used in the Nusselt number equation the same shown in Table 1 of the supplementary material?*

Response: We apologize for the confusion. H is the thickness of the working fluid layer which is same as height of Rayleigh Bénard convection cell. In Table 1, the first four parameters that all define the geometry of the experimental setup, so here we describe H as the height of Rayleigh Bénard convection cell. Therefore the parameter H used in the Nusselt number equation is the same as shown in Table 1 of the supplementary material. In the definition of Nusselt number we show in the paper $Nu = \dot{Q}/(\lambda \frac{\Delta T}{H})$, in which \dot{Q} is the heat flux with units of W/m^2 . We agree that \dot{Q} is not a good notation for the heat flux. Hence we now changed it to Q , which is routinely used in Rayleigh-Bénard convection system.

Comment 2: *Introduction, Page 2: The authors state that the power law dependence can be attributed to the interaction between the turbulent bulk and boundary layer. I think the authors mean turbulent bulk flow, right? Also, the authors indicate that Nusselt number scales with an effective scaling exponent β less than $1/3$. What scaling exponent are the authors referring to ?*

Response: Yes, the reviewer is correct. We mean the turbulent bulk flow. We now included this in the revised manuscript. The relation between Nusselt and Rayleigh has been studied extensively in classical Rayleigh-Bénard turbulence literature. Based on the experiments, theoretical analysis and numerical simulations, it is now clear that the relation between Nu and Ra follows roughly a power-law dependence with an effective scaling exponent β , i.e. $Nu = k Ra^\beta$. This exponent is a local exponent that also has a dependence with Ra . The value of the scaling exponent is determined by the competition between bulk flow and boundary layer contributions. For a wide range of Ra that are common in most engineering applications, the scaling exponent β is found to lie between $2/7$ and 0.33 (Niemela et al., 2000; Ahlers et al., 2009; Weiss and Ahlers, 2011). This is what we meant here. We have elaborated on this in the revised manuscript.

Comment 3: *Introduction: When taking into account Rayleigh number, do the authors consider the viscosity dependence on temperature?*

Response: Yes, this is taken into account. In the experiment, the temperature dependent viscosity of the working fluid is changing in between $0.000715 \text{ kg}/(\text{m s}) \sim 0.000981 \text{ kg}/(\text{m s})$.

Comment 4: *Results: When the authors indicate Nu_0 as the heat exchange due convective turbulence alone, are they referring to the case when there is no HFE-7000 in.*

Response: We refer to the case with a thin HFE-7000 layer for the single phase Nu_0 . When the bottom plate temperature T_b is below the critical temperature (boiling temperature) T_{cr} of HFE-7000, the thin layer of HFE-7000 is in liquid form. Because this case presents a direct and easy comparison between single phase and two phase. The thickness of the layer is small. If we use with no HFE-7000 as our reference case, our actual maximum heat transfer enhancement is reduced marginally from about 550% to 537% (see Figure. 1(b) of paper), i.e. still the five fold heat transfer increase. Here, we want to focus on Nu enhancement due to the boiling of HFE-7000 in the same system. We, therefore, compare Nu of the system in the two-phase situation as compared to that of the system in single-phase case (with HFE-7000).

Comment 5: *Results: From Fig. 2(e), it is not clear if the velocity shown in the graph is the resultant velocity in scalar form or just the magnitude of the vertical velocity. Please clarify.*

Response: The velocity shown here is the vertical velocity, with positive denoting upward travel of the biphasic riser elements, and negative denoting downward motion. The same convention holds for Fig 2(g). We now explain these in the revised paper.

Comment 6: *Results: What do the authors mean by the velocity being dynamically equilibrated? Is that the term they use when a terminal velocity is reached due to the combined effect of drag, buoyancy and gravity forces?*

Response: By dynamically equilibrated we mean the following: The system is a statistically stationary state. The flow is highly dynamic, but the collective motions of the flow and the global responses of the system are stable over time. The ratio of liquid and vapor phases change locally and dynamically, but substances move between the states at an equal rate, meaning there is no net change with time. We now have clarified this in the revised version of the manuscript.

Comment 7: *Results: The authors used HFE-7000 and claim that it is completely immiscible in water. However, according to the data provided by 3M (<https://multimedia.3m.com/mws/media/1213720/3m-novec-7000-engineered-fluidtds.pdf>) HFE-7000 has a miscibility or solubility value of 60 ppmw in water. Please elaborate.*

Response: We thank the reviewer for pointing this out. We apologize for the inaccurate term we use, and have corrected this in the revised manuscript. The solubility of 60 ppmw HFE-7000 in water can be converted to 43ppmv. The HFE-7000 liquid we add into the system is 1% of the cylindrical cell volume, if we take into account the solubility of HFE-7000 there is 99.57% HFE-7000 by volume which is still remains liquid and contributes to the biphasic catalytic particles and consequently the global heat transfer response. That is to say only 0.43% of the HFE-7000 liquid is lost into the water due to missibility. We expect this to not affect the system much. We have now given this detail more clearly in the revised paper and the Supplementary Material.

Comment 8: *Results: Even though the authors used a PIV system, they did not include rms velocity data to show the effect of catalytic particles on induced-liquid agitation and mixing explicitly and directly.*

Why? In fact, in the analysis, they used the theoretically estimated liquid velocity fluctuation (u') based on measured V_c and α . The authors should discuss on the merit of the approach followed instead of relying on direct measurements of rms.

Response: Unfortunately we did not find the rms value from PIV reliable because the flow is multiphase and has effects of active particles passing. This means, the rms runs the risk of including a bias from the sudden passage and exit of particles. The mean velocity is still accurate. Further, performing hot-film based techniques as done in past work cannot be used here because the flow is with strong temperature fluctuations. However, the collective velocity estimation is very reliable and repeatable. The analysis we performed here using V_c and α is based on results from many past papers which have carefully measured and modeled the liquid agitation multiphase flows (Risso, 2018). We also recently studied these and observed the model to be fairly accurate in other multiphase settings (Alm eras et al., 2019), which works well for the α range we have in the present experiments. We now clearly elaborate on these in the revised manuscript, and also highlight some approximations used and stress that the modeling is not based on direct velocity fluctuations, which can be highly challenging in such multiphase settings especially requiring thermal insulation and heat transfer.

Comment 9: *Results: The authors indicate that the biphasic particles smartly increase their overall density. What do they mean by smartly increasing the density?*

Response: We apologize for the confusion. We agree that the word ‘smartly’ is not accurate. In fact, what we mean here is that the HFE-7000 liquid boils to form vapor bubbles and the vapor bubbles will detach from the bottom plate and rise; at the meantime of rising through the bulk of the cell the vapor will condense into liquid, so the bubbles become biphasic with more and more liquid fraction and consequently more and more heavier until they reaches the top plate where they condense totally into liquid droplets. The liquid droplets will fall and return the bottom plate. Under the experimental condition, this phase-change cycle is spontaneous and self-sustained. We have deleted the word ‘smartly’ and explained more clearly in the revised manuscript.

Comment 10: *Results: In the equations for \dot{Q}_l the authors assume the rising plume to be completely cylindrical instead of having different shapes. Why? Please explain.*

Response: We don’t make the assumption of a cylindrical plume. In the equation for \dot{Q}_l , $\pi d^2/4$ measures the cross section area of the system, here the parameter d is the diameter of the cell not the plumes. For the biphasic kinematic contribution, we measure the vapor phase volume fraction α using the expansion vessel and we assume that the vapor bubbles (risers) distribute homogeneously inside the cell for a long time during the dynamically stable state. So when we calculate the vapor mass flux \dot{m}_v , we include the parameter α . We now describe this more clearly in the revised Supplementary Material.

Comment 11: *Results: Through modeling, the authors claim that the rising/settling biphasic particles are turbulent, instead of observing and quantifying the behavior directly through PIV. Why? Please explain.*

Response: We thank the reviewer for pointing out this. Direct measurement of the turbulence inside the insulated multiphase convection system is currently very challenging. However, we measured the Galileo number of the particles (Ern et al., 2012). The measured Ga number is larger than 1000, and it is found that Ga is similar to Re_p in this Galileo number regime (Horowitz and Williamson, 2010). This made us estimate that the particle wakes are turbulent (Horowitz and Williamson, 2010, 2008; Mathai et al., 2018). This has also been seen in our recent work (Alm eras et al., 2017). We now explain these in more detail in the revised paper.

Comment 12: *Supplementary Material, Experimental Setup: It appears that the authors used an im-*

immersion thermocouple to measure fluid temperature at a specific height. Was the thermocouple insulated or sheathed to avoid the so-called fin effect? How did the authors ensure that the temperature at the thermocouple junction was being measured accurately?

Response: The immersion thermistor as shown in Figure 1(a), is comprised of an insulation sheathed wire and the thermal resistance at the tip of the wire coated with very thin glass. The diameter of the thermistor head is about 0.38 mm, and the response time is 30 milliseconds in liquids which is fast enough for temperature measurements. The thermistor is inserted into the cell through a long thin steel tube and is bend roughly perpendicular to the tube in order to avoid the so-called fin effect. Figure 1(b) is the picture of the immersion thermistor and Figure 1(c) is a sketch for mechanical details. We have included the details in the revised Supplementary Material.

Figure 1: (a) Sketch for the way of how the immersion thermistor is inserted into the cell (NTC Temperature Sensor, rapid time response. Model GAG22K7MCD419, TE Connectivity Inc); (b) Photo of the real immersion thermistor; (c) Sketch for the immersion thermistor for mechanical details.

Comment 13: *Supplementary Material, Experimental Setup:* In Fig. 1, there is a typo in the figure itself. It should read *expansion vessel* (not *expansion vessel*).

Response: We are grateful to the reviewer for meticulous reading of the manuscript and supplementary material. We have corrected the typo.

Comment 14: *Supplementary Material, Experimental Setup:* What was the accuracy of the device used to measure α ?

Response: The method we use to get the volume fraction α is as follows. We measure the water surface height h inside the expansion vessel and according to h we can calculate α based on the conservation of mass and conservation of volume. The expansion vessel is with scales from which we can read the water surface height h . The scale on the expansion vessel has the minimum scale value 1mm, which corresponds to 1.7%

α . For each case, after the system reaches the finally statistically stable state and we then conduct about four-hour heat flux measurements. During the heat flux measurements, we monitor h from the expansion vessel scales every other 30 minutes and we in total obtain eight readings of h . On top of this, the minimum and maximum variation in α during the four hours measurement was 1.3% variance of α . So the value h we use to calculate α is the mean value of the eight readings. This has been included in the Methods part of the paper.

Comment 15: *Supplementary Material, Control and response parameters: It is not clear if the H used in the k_f equation is the same one as the one shown in Table 1.*

Response: H used in the k_f equation is the same one as the one shown in Table 1. Equation (6) in the Supplementary Material is the general definition of the Nusselt number, k_f is the heat transfer coefficient of pure conduction with unit $W/(m^2 \cdot K)$. The H is the thickness of the working fluid layer which is same as height of Rayleigh Bénard convection cell. We have included the details the revised Supplementary Material.

Comment 16: *Supplementary Material, Experiments: The authors claimed that only a small fraction of HFE-7000 liquid becomes vapor. Is the accuracy of the used device good enough to sustain or validate this claim?*

Response: Yes, only a small fraction of HFE-7000 liquid becomes vapor. In fact, the real evaporation fraction is small because the risers are biphasic and the real mass is mostly liquid (once the vapor bubbles detach from the bottom, they will condense as rising towards the top plate). This is further checked by the volume conservation technique. We use the expansion vessel and calculation to get α with measurement error about 3% which is precise enough to determine the bubble volume fraction. Within the boiling regime of the experiments, only 6% \sim 46% of the initial HFE-7000 liquid evaporates.

Comment 17: *Supplementary Material, Experiments: The use of the dye was used to compare mixing intensity, but at best, it can also be used to determine the extent of the mixing process, right? Please comment on the effectiveness of the dye from the analysis point of view.*

Response: The reviewer is correct. The dye injection was used mainly to quantify the mixing qualitatively. Quantitative measurements requires the usage of volume illumination and pre-calibration of Laser-induced fluorescence as we have done in Alméras et al. (2019). However this requires extensive setup modifications. We now explain that the mixing process reported here is to give an idea of the extent of the mixing process.

Comment 18: *Supplementary Material, Experiments: The authors claim that the biphasic system displays fast and chaotic mixing over a wide range of length scales. What range of length scales? Have the author considered a specific metric to quantify the fast and chaotic mixing process?*

Response: We thank the reviewer for deeper questions on the mixing. We notice that the typical structure size in single phase convection is limited to a large scale roll and the plumes. However, in biphasic case, we observe small scale mixed parcels of dye and also large scale, e.g. the large scale circulation, the plumes, biphasic particles and their induced fluid agitation. These different structures all contribute to the global heat transfer. We now measured the smallest size dye patches after 24 seconds, and compared this to single phase case. Figure 2 shows the histogram and the cumulative histogram of all kinds of scales of mixing structure areas A normalized by the vertical cross section area of the cell $A_0 = d \times H$, where d and H are the diameter and the height of the experimental cell respectively. Compared with single phase regime, the biphasic regime displays richer mixing structures, which indicates fast and chaotic mixing over a wide range of length scales. This detailed explanations have been reported in the revised Supplementary Material.

Figure 2: (a) Histogram for normalized single phase regime mixing structure area; (b) histogram for normalized biphasic regime mixing structure area; (c) cumulative histogram of the normalized structure areas for single phase regime and biphasic regime.

Comment 19: *Supplementary Material, Calculation of biphasic kinematics contribution: Equation 15 is based on the assumption that a cylindrical plume forms and can be used in the subsequent analysis. Please elaborate.*

Response: We don't make the assumption of a cylindrical plume. In the equation for \dot{Q}_l , $\pi d^2/4$ measures the cross section area of the system, here the parameter d is the diameter of the cell not the plumes. For the biphasic kinematic contribution, we measure the vapor phase volume fraction α using the expansion vessel and we assume that the vapor bubbles (risers) distribute homogeneously inside the cell for a long time during the dynamically stable state. So when we calculate the vapor mass flux \dot{m}_v , we include the parameter α .

We thank the reviewer for many detailed comments on the paper, which helped improve the manuscript. We have implemented all of the suggestions in the revised paper. We hope that with these changes and additions, the paper can now be given full recommendation for publication in Nature Communications.

References

- Ahlers, G., Grossmann, S., and Lohse, D. (2009). Heat transfer and large scale dynamics in turbulent Rayleigh-Bénard convection. *Rev. Mod. Phys.*, 81:503.
- Alm eras, E., Mathai, V., Sun, C., and Lohse, D. (2019). Mixing induced by a bubble swarm rising through incident turbulence. *International Journal of Multiphase Flow*.
- Alm eras, E., Mathai, V., Lohse, D., and Sun, C. (2017). Experimental investigation of the turbulence induced by a bubble swarm rising within incident turbulence. *Journal of Fluid Mechanics*, 825:1091–1112.
- Ern, P., Risso, F., Fabre, D., and Magnaudet, J. (2012). Wake-induced oscillatory paths of bodies freely rising or falling in fluids. *Annual Review of Fluid Mechanics*, 44:97–121.

- Horowitz, M. and Williamson, C. (2008). Critical mass and a new periodic four-ring vortex wake mode for freely rising and falling spheres. *Physics of Fluids*, 20(10):101701.
- Horowitz, M. and Williamson, C. H. (2010). Vortex-induced vibration of a rising and falling cylinder. *Journal of Fluid Mechanics*, 662:352–383.
- Mathai, V., Zhu, X., Sun, C., and Lohse, D. (2018). Flutter to tumble transition of buoyant spheres triggered by rotational inertia changes. *Nature communications*, 9(1):1792.
- Niemela, J., Skrbek, L., Sreenivasan, K. R., and Donnelly, R. (2000). Turbulent convection at very high Rayleigh numbers. *Nature*, 404:837–840.
- Risso, F. (2018). Agitation, mixing, and transfers induced by bubbles. *Annual Review of Fluid Mechanics*, 50:25–48.
- Weiss, S. and Ahlers, G. (2011). Turbulent rayleigh–bénard convection in a cylindrical container with aspect ratio $\gamma = 0.50$ and prandtl number $pr = 4.38$. *Journal of Fluid Mechanics*, 676:5–40.

May 17, 2019

1 Response to Referee 2

General comment: *The manuscript presents an experimental study of a Rayleigh-Bernard convection with water and a small addition of HFE 7000 liquid. Under different temperature of the bottom plate, the HFE 7000 vaporises differently and generates bubbles, which the authors call active particles. They show a 5-fold increase in heat transfer than conventional convective heat transfer. It is concluded that the increased heat transfer is due to both the kinematics of the bubble and the bubble induced turbulent flow. The experimental work is nicely done, with the flow visualization, bubble velocity calculation, dye mixing measurements, etc. The work should be reproducible giving the level of details provided.*

The idea of inducing turbulence/bubbles to increase the heat transfer rate has been spelled previously. Surprisingly, there are very few experiments with multiphase (two-fluid mixture) Rayleigh Bernard convection. Though the HFE7000 is a known thermal fluid, its application to the RBC system is new. The present work is novel and should be of interest to a wide range of audiences interested in heat exchange.

Overall, I think the paper is interesting and is suitable for publications in Nature Communications. Below are several questions to improve the presentation of the results.

Response: We thank the reviewer for the detailed reading of our manuscript and for his/her positive comments. Below we provide detailed answers to the questions raised by the reviewer. We also point out how we have revised the paper in accordance with the reviewer's suggestions.

Comment 1: *The authors emphasis in several places the self-sustainability of the particle (bubble) generated turbulence in the text. However, there are no experimental data showing, for example, long time behaviour of the system. Can the authors use some quantitative measure to show the sustainability?*

Response: The data we report is all performed under statistically stationary state after running the experiment for about ten hours. Below we show an additional figure as an example, Figure 1, showing the sustainability of the process. The standard deviation of the Nu time series is 0.028 which is accurate enough for the heat transfer measurement, the long time behaviour of the particle (bubble generated) system is stable and self-sustainable. We now included these descriptions in the revised manuscript.

Comment 2: *In Fig.3e, is the boundary of the shaded region (biphase kinematics) calculated using eq.15-17 from the SI? Since this is one of the main conclusions of the paper, it would be nice to have eq.15 in the main text.*

Response: Yes, the reviewer is right, the boundary of the shaded region (biphase kinematics) calculated using eq.15-17 from the SI. We thank the reviewer for pointing this out. We now included this in the revised manuscript for better readability.

We thank the referee for detailed comments on the paper, which helped to improve the manuscript. We

Figure 1: Nusselt number time series of statistically steady state of $T_b - T_{cr} = 5.27K$. The standard deviation of the Nu time series is 0.028, the long time behaviour of the particle (bubble generated) system is self-sustainable.

hope that with these changes and additions, the paper can now be given full recommendation for publication in Nature Communications.

References

May 17, 2019

1 Response to Referee 3

General comment: *This article reports on the experimental demonstration of a novel (or anyway not previously identified) class of multiphase turbulent transfer. This is the result of the addition of a small amount of heavy fluid to a classic convective cell containing water. The heavy fluid boils at relatively low temperature, producing buoyant “particles” that rise and settle following a dynamic condensation/evaporation equilibrium. The mixing induced by these particles produces a stunning five-fold increase of Nusselt number, compared to a classic turbulent thermal convection. The evidence provided by the authors is compelling and strongly supports their claims.*

The result is truly remarkable, with potential impact in a vast range of applications from nuclear to biomedical engineering, as highlighted by the authors. Therefore this work is expected to be of interest to a broad scientific audience. Moreover, it surfaces a rich and relatively unexplored dynamics of self-sustained particle-driven turbulent convection. Thus the work will also trigger strong interest from the specialized community in the area of turbulent multiphase flows. In the interest of clarity and completeness, I suggest the following points are addressed before publication:

Response: We thank the referee for the detailed reading of our manuscript and for his/her positive comments. Below we provide detailed answers to the questions raised by the referee. We also point out how we have revised the paper in accordance with the referee’s suggestions.

Comment 1: *The authors refer to a “critical temperature” T_{cr} , beyond which the heavy fluid activates. This is actually the boiling temperature of this fluid, but this is only clear in the supplementary material. It should be clarified in the main article.*

Response: We apologize for not explaining more and have stressed T_{cr} is the boiling point of HFE 7000 in the revised main article.

Comment 2: *Moreover, the “particles” of the heavy fluid are shown as “rising bubbles” and “falling droplets” in Figure 2, but one has to look carefully at such figure to realize that there is a complete phase transition involved. The authors need to be clearer in describing the steps of the process, illustrating how the fluid evaporates and condense, and clearly stating in which part of the dynamic process the fluid is gaseous/liquid.*

Response: We thank the reviewer for this point. The HFE-7000 liquid boils to form vapor bubbles and the vapor bubbles will detach from the bottom plate and rise; at the meantime of rising through the bulk of the cell the vapor will condense into liquid; so the bubbles become biphasic with more and more liquid fraction, and consequently more and more heavier until they reach the top plate, where they condense totally into liquid droplets. The liquid droplets will fall and return the bottom plate. In the revised paper, we have now added a short paragraph detailing the process.

Comment 3: *The five-fold increase of Nusselt number is impressive, and so is the twelve-fold increase*

in convection velocity scale. However, it is not obvious how those quantities might scale. The authors do provide interpretation/explanation on the processes leading to the enhancement of turbulent fluxes, but it is not evident how this can extrapolate to, for example, higher temperature differences. Is the linear behavior in Figure 1(b) expected to continue?

Response: In the experiment we keep the temperature difference ΔT between the top plate temperature T_t and bottom plate temperature T_b to be nearly constant (about 30 K), which leads to measurement protocol as follow. We change the top and bottom plate temperature by the same amount in the same direction, e.g. for one case $T_t = 5^\circ\text{C}$, $T_b = 35^\circ\text{C}$; for another case $T_t = 10^\circ\text{C}$, $T_b = 40^\circ\text{C}$, etc. For the classical Rayleigh Bénard convection the response parameter Nu and the control parameter Ra has a scaling relation $\text{Nu} = k\text{Ra}^{0.309}$ Niemela et al. (2000); Ahlers et al. (2009), using the data we obtained in the single phase regime from the experiment we can get the value of the prefactor k . For the classical Rayleigh Bénard convection we expect the heat transfer to continue the trend of $\text{Nu} = k\text{Ra}^{0.309}$ and thus we can predict the effective Rayleigh number for achieving a comparable highest heat flux enhancement through thermal plumes in the classical Rayleigh Bénard convection (see the inset of fig.3(e) in the main article). So for the given experimental setup, a comparable heat flux enhancement through thermal plumes would demand about sixty-five fold increase in the effective Ra, i.e. equivalent to an effective ΔT increase from 30 K to 1950 K. We have explained more details in the main article. We do not know whether the linear behavior in Figure 1(b) will continue, as the current analysis is based on the measured data in the present parameter regime. This certainly will be an interesting issue to be studied in future.

Comment 4: Figure 2(c) is supposed to show a state of high frequency and high amplitude temperature fluctuations, compared to the other panels in the same figure. But from the figure, I'd rather say that the fluctuations have intermediate frequency and amplitude, between those of panels (a) and (b). Maybe things would be clearer if the Supplementary Material included a Fourier analysis of those temperature fluctuations.

Response: Yes, the reviewer is correct. We apologize for the confusion. We perform the Fourier analysis of those temperature fluctuations as shown in figure 1. We can see from the figure 1(a) that in the single phase regime the temperature fluctuation frequency mostly distributes in the range between $0 \sim 0.3$ Hz but centralized around 0.016 Hz and the fluctuation amplitude is high; figure 1(b) is in the partially active regime, the frequency distributes mostly between the range $0 \sim 0.25$ Hz and is centralized around 0.06 Hz and the amplitude is the smallest among the three regimes; figure 1(c) is in the fully active regime, the frequency distributes mostly between the range $0 \sim 0.4$ Hz and is centralized around 0.004~0.036 Hz and around 0.125 Hz, while the amplitude is in between the single phase and the partially active regimes. We now clearly elaborate on these in the revised manuscript and the Supplementary Material.

Comment 5: I couldn't find the definition of the variable T_m in Figure 2.

Response: T_m is the mean temperature of the working fluid, and it is defined as $T_m = (T_t + T_b)/2$. We have included the details in the revised manuscript.

Comment 6: The authors rightfully stress the novelty of their findings. One of the few examples of regimes in which particles are driving turbulent thermal convection in a self-sustaining fashion is the work of Zamansky et al., *Phys Fluids* 26, 071701, 2014. Similarity and differences with respect to the present case should be mentioned.

Response: We thank the reviewer for pointing out this relevant reference. Indeed, our work has a close connection with the work by Zamansky et al. (2014). The self-sustaining fashion described in the work of Zamansky et al. (2014) is induced by the three-way coupled interactions between the transport modes associated with fluid flow, particles, and radiation, and their resulting radiation-induced turbulence, i.e. the response of the fluid to the forcing enhances the forcing itself. In our investigation, the self-sustaining is refer to the state that the HFE 7000 liquid can boil on the bottom plate to form a large cluster of bubbles,

Figure 1: Fourier analysis of temperature fluctuations.

then the bubbles can rise through the bulk region of the cell and reach the top plate; at the top plate, all the bubbles will condense into droplets; the droplets will settle down to the bottom plate which return to the non-boiling condition. Though details of two processes are quite different, they do have a close connection, i.e. making use of self-sustaining cycle. We have cited this paper in the main article and explained the connections in between.

Comment 7: *The phrase quiescent regime, used by the authors to refer to the regime before the activation of the heavy fluid, is contradictory with the fact that this really is a turbulent convection. A different wording is in order.*

Response: We thank the reviewer for pointing this out. We apologize for the misleading phrase we use to describe the HFE 7000 non-boiling regime during which the flow field is really a turbulent convection. We have change the phrase quiescent to ‘single phase regime’ in the revised manuscript.

We thank the referee for many detailed comments on the paper, which helped improve the manuscript. We have implemented all of the suggestions in the revised paper. We hope that with these changes and additions, the paper can now be given full recommendation for publication in Nature Communications.

References

- Ahlers, G., Grossmann, S., and Lohse, D. (2009). Heat transfer and large scale dynamics in turbulent Rayleigh-Bénard convection. *Rev. Mod. Phys.*, 81:503.
- Niemela, J., Skrbek, L., Sreenivasan, K. R., and Donnelly, R. (2000). Turbulent convection at very high

Rayleigh numbers. *Nature*, 404:837–840.

Zamansky, R., Coletti, F., Massot, M., and Mani, A. (2014). Radiation induces turbulence in particle-laden fluids. *Physics of Fluids*, 26(7):071701.

June 23, 2019

In summary, I enjoyed reading your paper and learning about your ground-breaking work. Thanks for addressing the initial review comments. I have found few additional but minor items that should be corrected, as follows:

1 Response to Reviewer 1

Comment 1: *In Supplementary Material, page 8, top of the page, where it reads: "..the long time behavior of the particle (bubble generated)..." there is a space between "(" and "bubble" Please delete the additional space.*

Response: Done.

Comment 2: *In Supplementary Material, page 12, near the bottom of the page, the sentence that start with "We now measured..." delete the word "now" since you are using past tense throughout the paragraph.*

Response: Done.

Comment 3: *In Supplementary Material, Figure 8(c), the legend show "biphasic regiem" instead of "biphasic regime" Please correct accordingly.*

Response: Done.

Comment 4: *Please check both documents (main manuscript and supplementary material) to make sure there are no typographical errors in them.*

Response: Done.